# Neurophysiological evidence of efference copies to inner speech

**Thomas J Whitford[1,2]\*, Bradley N Jack[1,2], Daniel Pearson[1,2], Oren Griffiths[1,2], David Luque[1,3], Anthony WF Harris[2,4], Kevin M Spencer[5,6], Mike E Le Pelley[1]**

[1]School of Psychology, University of New South Wales (UNSW Sydney), Sydney, Australia; [2]Brain Dynamics Centre, Westmead Institute for Medical Research, Sydney, Australia; [3]Department of Basic Psychology, University of Malaga, Malaga, Spain; [4]Discipline of Psychiatry, University of Sydney, Sydney, Australia; [5]Veterans Affairs Boston Healthcare System, Boston, United States; [6]Department of Psychiatry, Harvard Medical School, Boston, United States

**Abstract** Efference copies refer to internal duplicates of movement-producing neural signals. Their primary function is to predict, and often suppress, the sensory consequences of willed movements. Efference copies have been almost exclusively investigated in the context of overt movements. The current electrophysiological study employed a novel design to show that inner speech – the silent production of words in one's mind – is also associated with an efference copy. Participants produced an inner phoneme at a precisely specified time, at which an audible phoneme was concurrently presented. The production of the inner phoneme resulted in electrophysiological suppression, but only if the content of the inner phoneme matched the content of the audible phoneme. These results demonstrate that inner speech – a purely mental action – is associated with an efference copy with detailed auditory properties. These findings suggest that inner speech may ultimately reflect a special type of overt speech.
DOI: https://doi.org/10.7554/eLife.28197.001

\*For correspondence:
t.whitford@unsw.edu.au

## Introduction

Sensory attenuation – also known as self-suppression – refers to the phenomenon that self-generated sensations feel less salient, and evoke a smaller neurophysiological response, than externally-generated sensations which are physically identical (*Hughes et al., 2013*; *Cardoso-Leite et al., 2010*). Sensory attenuation is believed to result from the action of an internal forward model, or IFM (*Blakemore et al., 2000a*; *Wolpert and Miall, 1996*). According to this account, the sensory consequences of self-generated movements are predicted based on a copy of the outgoing motor command, known as an efference copy. These *predicted* sensations are compared to the *actual* sensations resulting from the movement, and the *difference* between predicted and actual sensation (i.e., the sensory discrepancy – *Wolpert and Miall, 1996*) is sent higher up the neuronal hierarchy for further processing (*Seth and Friston, 2016*). In the case of a self-generated movement, the internal prediction is able to account for, and 'explain away', much of the resulting sensation, which is why self-initiated sensations typically feel less salient, and evoke a smaller neurophysiological response, than externally-initiated sensations (*Blakemore et al., 1998*).

Sensory attenuation has been extensively studied in the context of overt speech production. Auditory stimuli elicit an electrophysiological brain response (the auditory-evoked potential) with a characteristic N1 component. The amplitude of this component is known to be sensitive to sound intensity; i.e., loud sounds evoke larger N1 amplitudes than soft sounds (*Näätänen and Picton, 1987*; *Hegerl and Juckel, 1993*). Numerous electroencephalographic (EEG) and magnetoencephalographic (MEG) studies have found that self-generated vocalizations elicit an N1 component (M100

**eLife digest** As you read this text, the chances are you can hear your own inner voice narrating the words. You may hear your inner voice again when silently considering what to have for lunch, or imagining how a phone conversation this afternoon will play out. Estimates suggest that we spend at least a quarter of our lives listening to our own inner speech. But to what extent does the brain distinguish between inner speech and the sounds we produce when we speak out loud?

Listening to a recording of your own voice activates the brain more than hearing yourself speak out loud. This is because when the brain sends instructions to the lips, tongue, and vocal cords telling them to move, it also makes a copy of these instructions. This is known as an efference copy, and it enables regions of the brain that process sounds to predict what they are about to hear. When the actual sounds match those predicted – as when you hear yourself speak out loud – the brain's sound-processing regions dampen down their responses.

But does the inner speech in our heads also generate an efference copy? To find out, Whitford et al. tracked the brain activity of healthy volunteers as they listened to speech sounds through headphones. While listening to the sounds, the volunteers had to produce either the same speech sound or a different speech sound inside their heads. A specific type of brain activity decreased whenever the inner speech sound matched the external speech sound. This decrease did not occur when the two sounds were different. This suggests that the brain produces an efference copy for inner speech similar to that for external speech.

These findings could ultimately benefit people who suffer from psychotic symptoms, for example as part of schizophrenia. Symptoms such as hearing voices are thought to reflect problems with producing and interpreting inner speech. The technique that Whitford et al. have developed will enable us to test this long-held but hitherto untestable idea. The results should increase our understanding of these symptoms and may eventually lead to new treatments.
DOI: https://doi.org/10.7554/eLife.28197.002

in the MEG) of smaller amplitude than the N1 component elicited when passively listening to the same sounds (*Ford et al., 2007a*; *Curio et al., 2000*; *Heinks-Maldonado et al., 2005*; *Oestreich et al., 2015*; *Houde et al., 2002*). This phenomenon has been dubbed N1-suppression, and it suggests that self-generated sounds are processed as though they were physically softer than externally-generated sounds, reflecting the action of an IFM (*Greenlee et al., 2011*; *Heinks-Maldonado et al., 2006*). The suggestion that an IFM is responsible for sensory attenuation to overt speech is bolstered by the finding that experimentally altering auditory feedback (e.g., by pitch-shifting or delaying an individual's voice, such that auditory sensations do not match the predictions of the IFM) reduces the amount of N1-suppression (*Heinks-Maldonado et al., 2006*; *Behroozmand and Larson, 2011*; *Behroozmand et al., 2011*; *Aliu et al., 2009*).

The central aim of the present study is to explore whether N1-suppression, which has consistently been observed in response to overt speech, also occurs in response to *inner speech*, which is a purely mental action. Inner speech – also known as covert speech, imagined speech, or verbal thoughts – refers to the silent production of words in one's mind (*Perrone-Bertolotti et al., 2014*; *Alderson-Day and Fernyhough, 2015*). Inner speech is one of the most pervasive and ubiquitous of human activities; it has been estimated that most people spend at least a quarter of their lives engaged in inner speech (*Heavey and Hurlburt, 2008*). An influential account of inner speech suggests that it ultimately reflects a special case of overt speech in which the articulator organs (e.g., mouth, tongue, larynx) do not actually move; that is, inner speech is conceptualized as *'a kind of action'* (*Jones and Fernyhough, 2007*, p.396 – see also *Feinberg, 1978*; *Pickering and Garrod, 2013*; *Oppenheim and Dell, 2010*). Support for this idea has been provided by studies showing that inner speech activates similar brain regions to overt speech, including audition and language-related perceptual areas and supplementary motor areas, but does not typically activate primary motor cortex (*Palmer et al., 2001*; *Zatorre et al., 1996*; *Aleman et al., 2005*; *Shuster and Lemieux, 2005*). While previous data suggest that inner and overt speech share neural generators, relatively few neurophysiological studies have explored the extent to which these two processes are *functionally* equivalent. If inner speech is indeed a special case of overt speech – *'a kind of action'* –

then it would also be expected to have an associated IFM (*Tian and Poeppel, 2010* – see also *Feinberg, 1978*; *Numminen and Curio, 1999*; *Whitford et al., 2012*; *Ford et al., 2001a*).

A significant challenge in determining the existence (or otherwise) of an IFM to inner speech is that inner speech does not elicit a measurable auditory-evoked potential, which means that N1-suppression to inner speech cannot be determined directly using current methodologies. However, the existence of an IFM may be inferred based on how the production of *inner* speech suppresses the brain's electrophysiological response to *overt* speech (*Tian and Poeppel, 2010*; *Tian and Poeppel, 2015*; *Tian and Poeppel, 2013*; *Tian and Poeppel, 2012*). A critical feature of the IFM associated with overt speech is that the efference copy is (a) time-locked to the onset of the action, and (b) contains specific predictions as to the expected sensory consequences of that action (i.e., is *content-specific* – *Wolpert et al., 1995*). Correspondingly, if inner speech were, in fact, associated with an IFM, then its associated efference copy would be expected to be: (a) time-locked to the onset of the inner speech, and (b) contain information as to the specific content of the inner speech. In this case, engaging in inner speech would be expected to result in maximal N1-suppression to overt speech in the case where two conditions were met: (1) the external sound was presented at precisely the same time as the inner speech was produced, and (2) the content of the external sound matched the content of the inner speech.

The present study introduces a new experimental procedure that allowed us to test whether inner speech produces N1-suppression to audible speech in the absence of any overt motor action. In this protocol, participants were instructed to produce a single phoneme in inner speech at a specific time, which was designated by means of a precise visual cue. At the same time, an audible phoneme was presented in participants' headphones; the audible phoneme could be either the same as (Match condition) or different from (Mismatch condition) the inner phoneme. In the Passive condition, participants were instructed not to produce an inner phoneme. The results indicated that inner speech resulted in N1-suppression, *but only if* the content of the inner phoneme matched the content of the audible phoneme. These results suggest that inner speech production is associated with a time-locked and content-specific internal forward model, similar to the one that operates in the production of overt speech. Furthermore, these results suggests that inner speech, by itself, is able to elicit an efference copy and cause sensory attenuation, even in the absence of an overt motor action.

## Results

### Inner speech experiment

On each trial of the experiment, participants watched a short animation of approximately 5 s duration. As illustrated in *Figure 1a*, the animation depicted a red vertical line (the 'fixation' line) that remained in a fixed location in the middle of the screen. This fixation line was overlaid upon a thick green horizontal bar (the 'ticker tape'). A second, green, vertical line (the 'trigger' line), which was embedded in the ticker tape, was initially presented at the far right-hand side of the screen. At the start of each trial, participants fixated their eyes on the fixation line. After an interval of 1–2 s, the green ticker tape and the green trigger line began to move leftwards across the screen towards, and ultimately beyond, the stationary fixation line. At the exact time at which the trigger line intersected the fixation line – the 'sound-time' – an *audible phoneme* was delivered to participants' headphones (*Figure 1c*). The audible phoneme was a recording of a male speaker producing either the phoneme /BA/ or the phoneme /BI/. (Note: for clarity, audible phonemes are capitalized in text while inner phonemes are written in lower case. Our justification for choosing these two audible phonemes in particular is presented below). Participants were instructed to generate an inner phoneme at exactly the moment the fixation line intersected the trigger line (i.e., at the sound-time). The experimental manipulation was the content of the inner phoneme that participants were instructed to produce. There were three different types of trial blocks. In the first type of trial block, participants were asked to produce the inner phoneme /ba/ at the sound-time. The second type of trial block was identical, except that participants were asked to produce the inner phoneme /bi/. On each trial, participants were asked to imagine themselves moving their articulator organs (i.e., mouth, tongue, larynx, etc.) and vocalizing the inner phoneme, but not to actually make any movements. In the third type of trial block, participants were not instructed to produce an inner phoneme,

**Figure 1.** A schematic of the experimental protocol. Participants were instructed to fixate their eyes on the central red fixation line (Panel A). After a delay (1–2 s), the green trigger line, which was presented on the far right-hand side of the screen, and visible in participants' peripheral vision, began to move smoothly across the screen in a leftwards direction at a speed of 6.5°/s (Panel B), such that after 3.75 s the trigger line overlapped with the fixation line. At this precise moment, dubbed the 'sound-time', two events occurred simultaneously (Panel C). Firstly, the participant was asked to imagine themselves producing a pre-defined phoneme in inner speech (either /ba/ or /bi/ or no inner phoneme). Secondly, an audible phoneme (either /BA/ or /BI/), produced by a male speaker, was delivered to the participant's headphones. In Match trials (Panel D, top, blue), the inner phoneme was congruent with the audible phoneme (e.g., inner phoneme: /ba/; audible phoneme: /BA/). In Mismatch trials (Panel D, middle, red), the inner phoneme was incongruent with the audible phoneme (e.g., inner phoneme: /bi/; audible phoneme: /BA/). In Passive trials (Panel D, bottom, black), the participant did not produce an inner phoneme. Following the sound-time, the trigger line continued to move past the fixation line for an additional 1 s. The trial was then complete and the participant was asked to rate how successfully they managed to follow the instructions on that trial, on a scale from 1 (Not at all successful) to 5 (Completely successful).

DOI: https://doi.org/10.7554/eLife.28197.003

but were instructed to simply listen to the sounds. Following the sound-time, the trigger line continued to move for an additional 1 s before a text-box was displayed and participants were asked to rate how successfully they followed the instructions on the trial.

The data were parsed into three discrete trial-types, which were analyzed as separate conditions: (1) *Match* trials, in which the inner phoneme matched the audible phoneme (i.e., inner phoneme: /ba/; audible phoneme: /BA/ OR inner phoneme: /bi/; audible phoneme: /BI/), (2) *Mismatch* trials, in which the imagined phoneme did not match the audible phoneme (i.e., inner phoneme: /ba/; audible phoneme: /BI/ OR inner phoneme: /bi/; audible phoneme: /BA/, and (3) *Passive* trials, in which the participant was not instructed to imagine a phoneme (i.e., inner phoneme: none; audible phoneme: /BA/ OR inner phoneme: none; audible phoneme: /BI/).

Following pre-processing (see *EEG Processing and Analysis* for full details), EEG data epochs (time-locked to the onset of the audible phoneme) were averaged separately for each of the three conditions (Match, Mismatch, Passive). The dependent variable was the amplitude of the N1 component of the auditory-evoked potential elicited by the auditory phoneme.

The N1 peak was identified on each individual participant's average waveform for each of the three conditions. *Figure 2* shows the auditory-evoked potentials averaged across electrodes FCz, Fz, and Cz, as these were the electrodes at which N1 was maximal (see *Figure 2a*, voltage maps). *Figure 2b* shows a box-and-whiskers plot of raw N1 amplitudes for each condition (Match, Mismatch, Passive). Given that this experiment used a repeated measures design, *Figure 2c* shows a scatterplot of the magnitude of the within-subjects differences between conditions, which constitute the critical contrasts. These difference scores were approximately normally distributed with no clear outliers. Repeated measures ANOVA revealed a significant main effect of *Condition* ($F_{(2,82)} = 4.21$, $p = 0.018$, $\eta_p^2 = 0.09$) on the amplitude of the N1 peak. Analysis of simple effects revealed that N1-amplitude in the Match condition was significantly smaller than both the Mismatch ($t(41) = 2.54$, $p = 0.015$, $dz = 0.39$, CI(95%) = [0.187, 1.649]) and Passive ($t(41) = 2.77$, $p = 0.008$, $dz = 0.43$, CI(95%) = [0.278, 1.776]) conditions (*Figure 2c*). There was no difference in N1-amplitude between the Mismatch and Passive conditions ($t(41) = 0.26$, $p = 0.800$, $dz = 0.04$, CI(95%) = [−0.758, 0.977]).

As can be seen in *Figure 2*, while the topographies exhibited a fronto-central negativity in all three conditions, centered on electrode FCz, there was a hint of a leftward shift in the scalp distribution in the Match condition. Thus, in order to ensure the stability of the results, we performed a supplementary analysis with an expanded set of nine electrodes: specifically, Fz, FCz, Cz, F1, FC1, C1, F2, FC2, and C2. The pattern of results was identical to when the analysis was restricted to the three

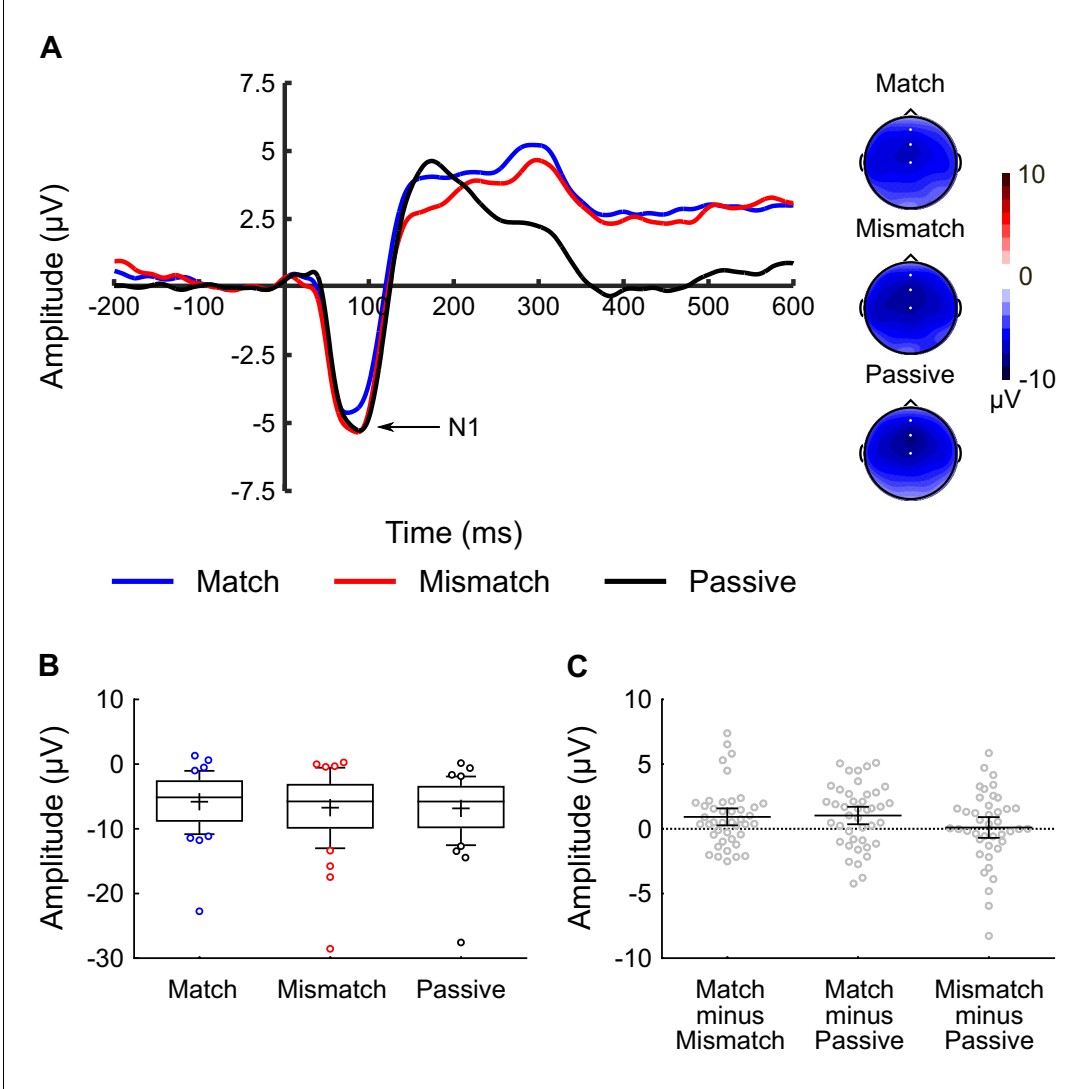

**Figure 2.** Inner speech experiment: N1 component analysis. (**A**) Waveforms showing the auditory-evoked potentials elicited by the audible phonemes in the Match condition (blue line), Mismatch condition (red line) and Passive condition (black line). The N1-component is labelled; the waveforms were averaged across electrodes FCz, Fz, and Cz, as these were the electrodes at which the N1 component was maximal. The waveforms are shown collapsed across audible phoneme (audible /BA/ and /BI/), and the waveforms for the Match and Mismatch conditions are shown collapsed across inner phoneme (inner /ba/ and /bi/). Voltage maps are plotted separately for each condition; white dots illustrate the electrodes used in the analysis. (**B**) Box-and-whiskers plots showing the amplitude of the N1 component elicited by the audible phonemes in the Match, Mismatch and Passive conditions. The edges of the boxes represent the top and bottom quartiles, the horizontal stripe represents the median, the cross represents the mean, the whiskers represent the 9th and 91st percentiles, and the colored dots represent the participants whose data fell outside the range defined by the whiskers. (**C**) Scatterplots showing the within-subjects difference scores (in terms of N1-amplitude) for the three contrasts-of-interest in the inner speech experiment; namely Match minus Mismatch, Match minus Passive, and Mismatch minus Passive. These difference scores were approximately normally distributed with no clear outliers. Each dot represents a single participant's difference score. The horizontal bars represent the mean, and the error bars represent the 95% confidence interval.

DOI: https://doi.org/10.7554/eLife.28197.004

The following source data is available for figure 2:

**Source data 1.** Inner Speech Experiment - N1 amplitude and latency data.
DOI: https://doi.org/10.7554/eLife.28197.005

midline electrodes. Specifically, the main effect of *Condition* remained significant (F(2,82) = 3.61, p = 0.031, $\eta_p^2$ = 0.08), as did the difference between the Match and Mismatch conditions (t(41) = 2.35, p = 0.024, dz = 0.36, CI(95%) = [0.122, 1.607]) and the Match and Passive conditions (t(41) = 2.57, p = 0.014, dz = 0.40, CI(95%) = [0.193, 1.624]). The difference between the Mismatch and Passive conditions remained non-significant (t(41) = 0.11, p = 0.916, dz = 0.02, CI(95%) = [−0.889, 0.800]). The Condition × Electrode interaction was also not significant (F(16,656) = 1.18, p = 0.323, $\eta_p^2$ = 0.03).

There was also a main effect of *Condition* on the latency of the N1 peak (F(2,82) = 5.03, p = 0.016, $\eta_p^2$ = 0.11. Analysis of the simple effects revealed that the N1-peak occurred significantly earlier in the Match condition compared to the Passive condition (t(41) = 3.15, p = 0.003, dz = 0.49, CI(95%) = [−6.927,−1.518]). There was no significant difference in N1-latency between the Mismatch and Passive conditions (t(41) = 1.76, p = 0.087, dz = 0.27, CI(95%) = [−6.298, 0.441]), nor between the Match and Mismatch conditions (t(41) = 1.29, p = 0.204, dz = 0.20, CI(95%) = [−3.316, 0.729]).

A visual inspection of *Figure 2* also suggested between-condition differences in the P2 (150–190 ms) and P3 (250–310 ms) components of the auditory-evoked potential. While these components were not directly relevant to our hypotheses, for completeness the data and analyses for these components are presented below.

The P2 component occurred around 150–190 ms post-sound (see *Figure 3a*), while the P3 component occurred around 250–310 ms post-sound (see *Figure 4a*). However, not all three conditions generated a distinct peak for the P2 and P3 components. Specifically, the Match and Mismatch conditions did not elicit a distinct P2, whereas the Passive condition did not exhibit a distinct P3. This meant that (unlike for analysis of the N1) it was not possible to use a peak-detection approach for these components. Instead, time-windows were identified for the P2 (150–190 ms) and P3 (250–310 ms) components, and the average voltage within these time-windows were analyzed (see *EEG Processing and Analysis* for more detail).

*Figure 3a* shows the average waveforms for the P2 component, averaged across the Cz, FCz, and CPz electrodes, and *Figure 3B* shows a box-and-whiskers plot of raw P2 amplitudes for each condition. One-way ANOVA revealed a main effect of *Condition* (F(2,82) = 6.60, p = 0.006, $\eta_p^2$ = 0.14). Analysis of the simple effects revealed that amplitude of the P2 component was significantly smaller in the Mismatch condition, relative to both the Match (t(41) = 3.54, p = 0.001, dz = 0.55, CI(95%) = [0.555, 2.028]) and Passive (t(41) = 3.21, p = 0.003, dz = 0.50, CI(95%) = [−3.549, −0.810]) conditions (*Figure 3C*). The difference between the Match and Passive conditions was not significant (t(41) = 1.26, p = 0.216, dz = 0.19, CI(95%) = [−0.540, 2.316]).

*Figure 4a* shows the average waveforms for the P3 component, averaged across the CPz, Cz, and Pz electrodes, and *Figure 4B* shows a box-and-whiskers plot of raw P3 amplitudes for each condition. ANOVA revealed a main effect of *Condition* (F(2,82) = 5.86, p = 0.004, $\eta_p^2$ = 0.13). Analysis of the simple effects revealed that the amplitude of the P3 component was significantly larger in the Match condition relative to both the Mismatch (t(41) = 2.23, p = 0.032, dz = 0.34, CI(95%) = [0.117, 2.433]) and Passive (t(41) = 3.26, p = 0.002, dz = 0.50, CI(95%) = [0.813, 3.444]) conditions (*Figure 4c*). There was no significant difference between the Passive and Mismatch conditions (t(41) = 1.31, p = 0.196, dz = 0.20, CI(95%) = [−0.458, 2.165]).

## Overt speech experiment

To provide a point of reference with the inner speech experiment, we also conducted a follow-up 'overt speech' experiment. The overt speech experiment had an identical experimental procedure to the inner speech experiment, except that participants were instructed to overtly – as opposed to covertly – vocalize the phonemes at the sound-time. Just as in the inner speech experiment, an audible phoneme (i.e., /BA/ or /BI/) was delivered to participants' headphones at the sound-time. Participants were instructed to vocalize the overt phonemes softly, so as to minimize the amount of bone conduction of the sound to the ear. An additional 'Motor-Control' condition was also included in which participants overtly vocalized the phonemes at the sound-time, but no audible phoneme was delivered. The purpose of this condition was to allow us to identify and correct for the electrophysiological activity generated by the motor act of producing the overt phoneme per se. This was done by subtracting participants' activity in the motor-only condition from their waveforms in the active conditions (i.e., Match and Mismatch), as is common in sensory attenuation studies which compare

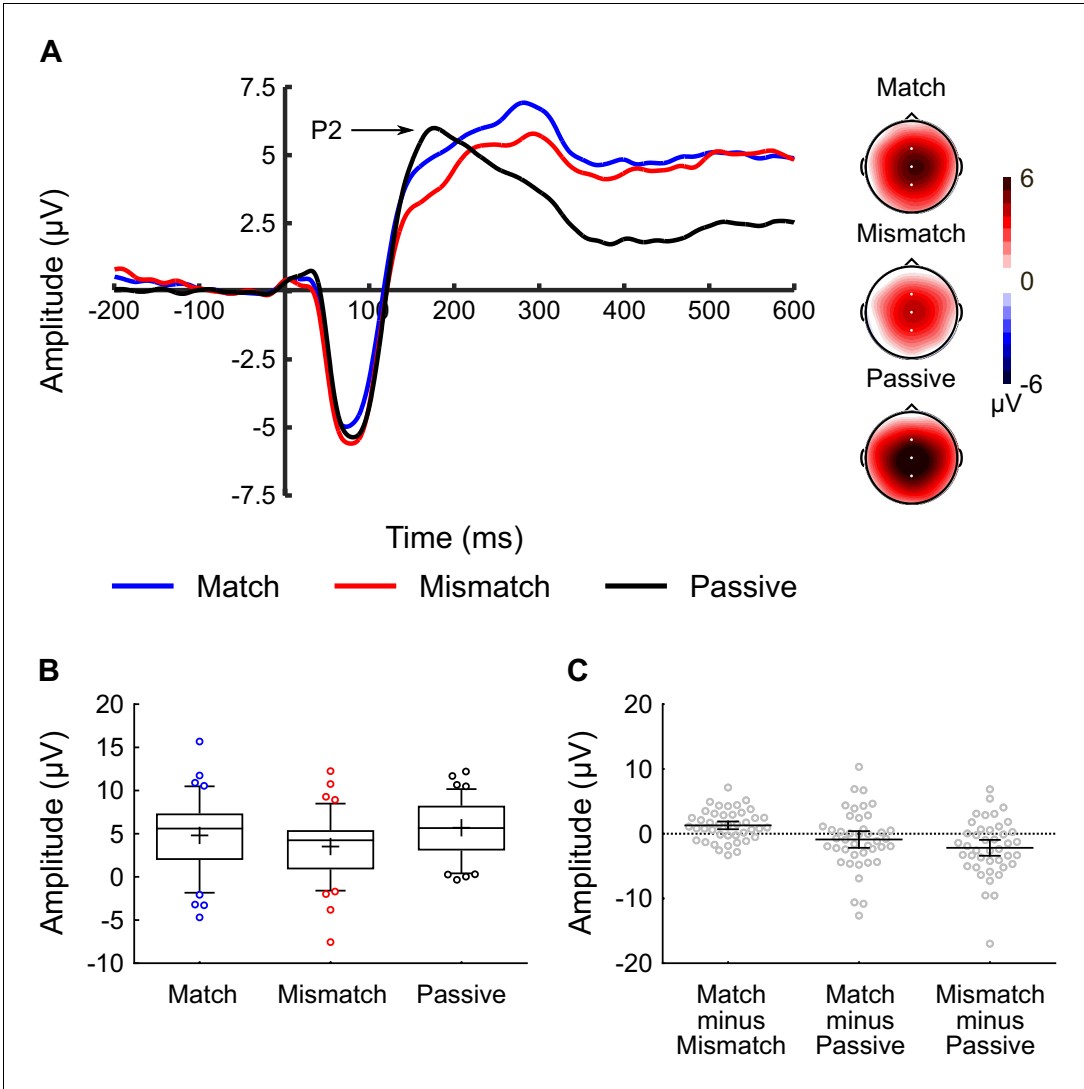

**Figure 3.** Inner speech experiment: P2 component analysis. (**A**) Waveforms showing the auditory-evoked potentials elicited by the audible phonemes in the Match condition (blue line), Mismatch condition (red line), and Passive condition (black line). The P2-component is labelled; P2 amplitude was calculated as the average voltage in the 150–190 ms time-window. The waveforms were averaged across electrodes Cz, FCz, and CPz, as these were the electrodes at which the P2 component was maximal. Voltage maps are plotted separately for each condition; white dots illustrate the electrodes used in the analysis. (**B**) Box-and-whiskers plots showing the amplitude of the P2 component elicited by the audible phonemes in the Match, Mismatch, and Passive conditions. The edges of the boxes represent the top and bottom quartiles, the horizontal stripe represents the median, the cross represents the mean, the whiskers represent the 9th and 91st percentiles, and the colored dots represent the participants whose raw data fell outside the range defined by the whiskers. (**C**) Scatterplots showing the within-subjects difference scores (in terms of P2-amplitude) for the three contrasts-of-interest in the inner speech experiment; namely Match minus Mismatch, Match minus Passive, and Mismatch minus Passive. These difference scores were approximately normally distributed with no clear outliers. Each dot represents a single participant's difference score. The horizontal bars represent the mean, and the error bars represent the 95% confidence interval.

DOI: https://doi.org/10.7554/eLife.28197.006

The following source data is available for figure 3:

**Source data 1.** Inner Speech Experiment - P2 amplitude data.
DOI: https://doi.org/10.7554/eLife.28197.007

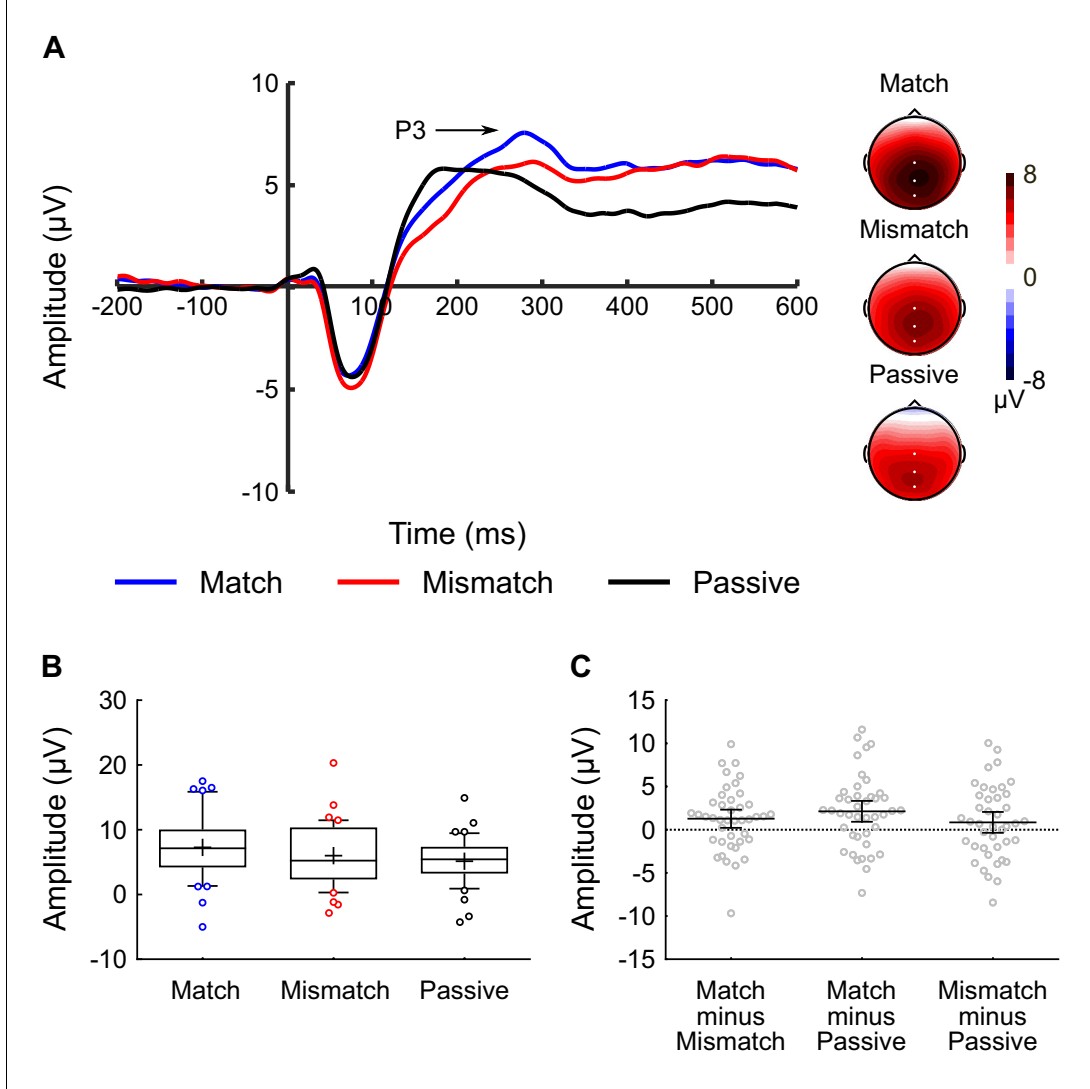

**Figure 4.** Inner speech experiment: P3 component analysis. (**A**) Waveforms showing the auditory-evoked potentials elicited by the audible phonemes in the Match condition (blue line), Mismatch condition (red line), and Passive condition (black line). The P3-component is labelled; P3 amplitude was calculated as the average voltage in the 250–310 ms time-window. The waveforms were averaged across electrodes CPz, Cz, and Pz, as these were the electrodes at which the P3 component was maximal. Voltage maps are plotted separately for each condition; white dots illustrate the electrodes used in the analysis. (**B**) Box-and-whiskers plots showing the amplitude of the P3 component elicited by the audible phonemes in the Match, Mismatch, and Passive conditions. The edges of the boxes represent the top and bottom quartiles, the horizontal stripe represents the median, the cross represents the mean, the whiskers represent the 9th and 91st percentiles, and the colored dots represent the participants whose raw data fell outside the range defined by the whiskers. (**C**) Scatterplots showing the within-subjects difference scores (in terms of P3-amplitude) for the three contrasts-of-interest in the inner speech experiment; namely Match minus Mismatch, Match minus Passive, and Mismatch minus Passive. These difference scores were approximately normally distributed with no clear outliers. Each dot represents a single participant's difference score. The horizontal bars represent the mean, and the error bars represent the 95% confidence interval.

DOI: https://doi.org/10.7554/eLife.28197.008

The following source data is available for figure 4:

**Source data 1.** Inner Speech Experiment - P3 amplitude data.

DOI: https://doi.org/10.7554/eLife.28197.009

motor-active and motor-passive conditions (*Ford et al., 2014*). The order of the conditions was randomized for each participant, with the caveat that the Motor-Control condition was always run last.

Thirty individuals participated in the overt speech experiment. Participants' mean age was 25.0 years (SD = 6.0) and 20 were female. Thirty-three participants were originally recruited for the study, however three participants generated ≤60 usable epochs in one or more conditions (based on the exclusion criteria described for the inner speech experiment) and were excluded from further analysis. *Figure 5* shows the auditory-evoked potentials averaged across electrodes FCz, Fz, and Cz for the uncorrected (*Figure 5a*) and motor-corrected data (*Figure 5b*). Voltage-maps are presented for the motor-corrected data. Raw N1 amplitudes for each motor-corrected condition are presented as box-and-whiskers plots in *Figure 5c*, and *Figure 5d* shows scatterplots of the (within-subjects) difference scores between the conditions.

For the uncorrected data, repeated-measures ANOVA revealed a significant main effect of *Condition* (F(2,58) = 10.99, p < 0.001, $\eta_p^2$ = 0.28) on the amplitude of the N1 peak. Critically, analysis of simple effects revealed that N1-amplitude in the Match condition was significantly smaller than the Mismatch condition (t(29) = 2.61, p = 0.014, dz = 0.48, CI(95%) = [0.472, 3.897]), consistent with the results of the inner speech experiment. However, contrary to the results of the inner speech experiment, N1-amplitude in the Passive condition was significantly smaller than both the Match (t(29) = 2.63, p = 0.013, dz = 0.48, CI(95%) = [0.646, 5.137]) and Mismatch (t(29) = 3.97, p < 0.001, dz = 0.72, CI(95%) = [2.461, 7.691]) conditions in the overt speech experiment (see *Figure 5a*).

The pattern of results was identical for the motor-corrected data. Repeated-measures ANOVA revealed a significant main effect of *Condition* (F(2,58) = 8.43, p = 0.001, $\eta_p^2$ = 0.23). Analysis of simple effects revealed that N1-amplitude in the Match condition was significantly smaller than the Mismatch condition (t(29) = 2.46, p = 0.020, dz = 0.45, CI(95%) = [0.384, 4.190]), and that N1-amplitude in the Passive condition was significantly smaller than both the Match (t(29) = 2.20, p = 0.036, dz = 0.40, CI(95%) = [0.150, 4.243]) and Mismatch (t(29) = 3.43, p = 0.002, dz = 0.63, CI(95%) = [1.808, 7.159]) conditions (see *Figure 5b, c and d*).

## Selecting the audible phonemes for the inner and overt speech experiments

In order to select which two audible phonemes would be presented to participants in the inner and overt speech experiments, we presented nine phonemes to 10 participants (age = 18.7 years, SD = 1.1; seven female) while they listened passively. Each phoneme was presented 90 times, and the presentation order was randomized. The nine phonemes were: /BA/, /BI/, /DA/, /DI/, /GA/, /KI/, /PA/, /PI/, and /TI/. Each phoneme was ~200 ms in duration, presented at ~70 dB SPL, and was produced by the same male speaker.

Waveforms showing the auditory-evoked potentials elicited by the nine phonemes are presented in *Figure 6*. The waveforms are shown collapsed across electrodes FCz, Cz, and Fz.

Of the nine different phonemes, /BA/ and /BI/ were judged to be most similar in terms of their amplitude and overall shape, and hence these phonemes were chosen for use as the audible phonemes in both the inner and overt speech experiments.

## Discussion

The present study used a novel experimental protocol to demonstrate that the production of an inner phoneme resulted in sensory attenuation of the auditory-evoked potential elicited by a simultaneously-presented audible phoneme, in the absence of any overt motor action. Crucially, the production of inner speech did not result in equal sensory attenuation to all sounds; sensory attenuation was dependent on the content of the inner phoneme matching the content of the audible phoneme. These results suggest that inner speech production is associated with a time-locked and content-specific internal forward model, similar to the one believed to operate in the production of overt speech (*Hickok, 2012*; *Tourville and Guenther, 2011*; *Hickok and Poeppel, 2004*; *Houde et al., 2013*). In short, the results of this study suggest that inner speech alone is able to elicit an efference copy and cause sensory attenuation of audible sounds.

The key finding of the present study was that the production of inner speech, by itself, led to N1-suppression to an audible sound. N1-suppression has been reported many times previously in response to overt speech (*Ford et al., 2007a*; *Curio et al., 2000*; *Heinks-Maldonado et al., 2005*;

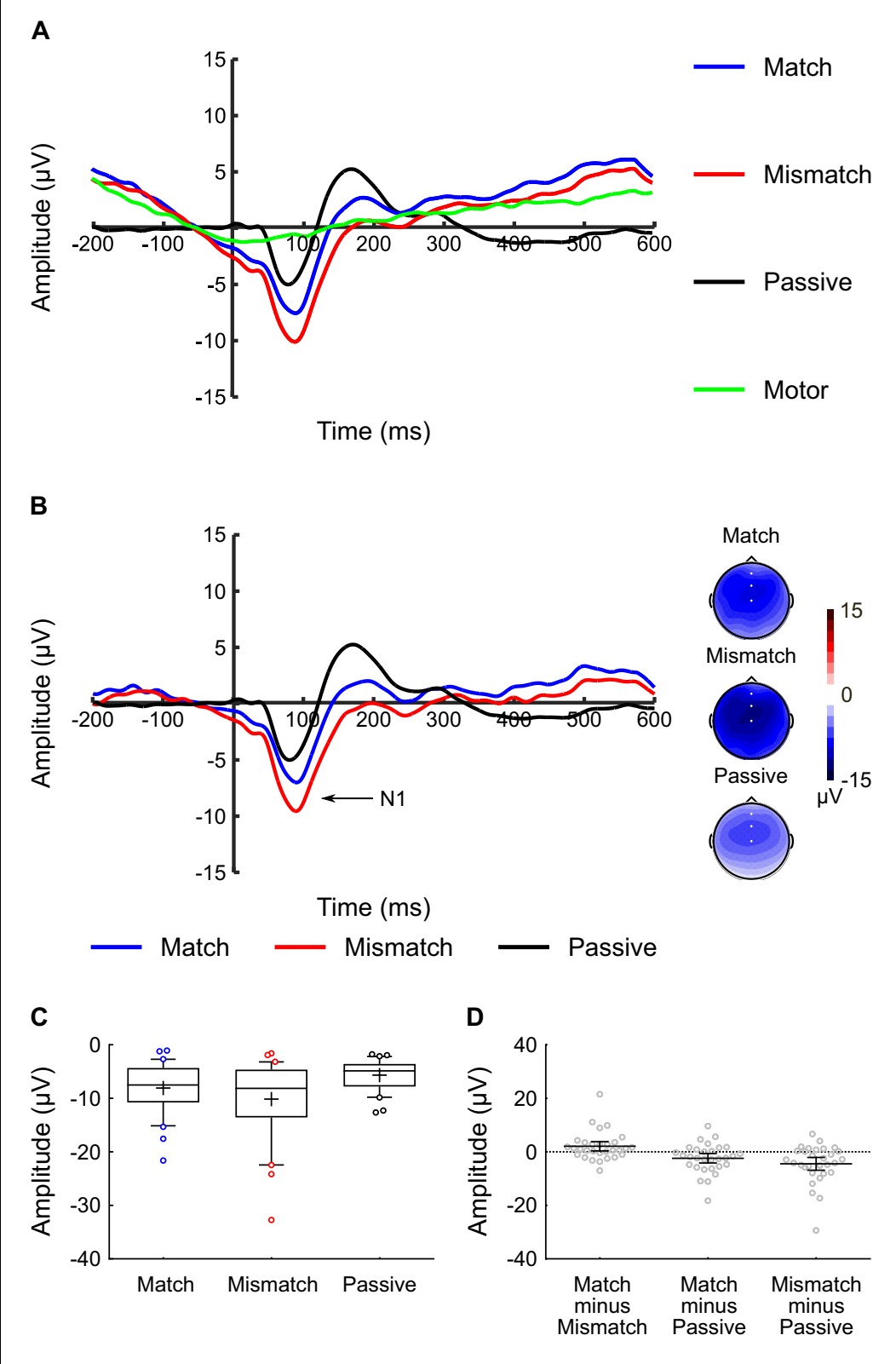

**Figure 5.** Overt speech experiment: N1 component analysis. The experimental protocol for the overt speech experiment was identical to the inner speech experiment except that participants were required to overtly (as opposed to covertly) vocalize the phoneme at the sound time. (**A**) Uncorrected waveforms showing the auditory-evoked potentials elicited by the audible phonemes in the Match condition (blue line), Mismatch condition (red

*Figure 5 continued*

line), and Passive condition (black line) in the overt speech experiment. The waveform for the motor-control condition is also shown (green line: in this condition participants overtly vocalized a phoneme at the sound-time, but no audible phoneme was delivered). The N1-component is labelled; the waveforms were averaged across electrodes FCz, Fz, and Cz, as these were the electrodes at which the N1 component was maximal. The waveforms are shown collapsed across audible phoneme (audible /BA/ and /BI/), and the waveforms for the Match, Mismatch, and Motor Control conditions are shown collapsed across vocalized phoneme (overt /ba/ and /bi/). (B) Motor-corrected waveforms showing the auditory-evoked potentials elicited by the audible phonemes in the Match condition (blue line), Mismatch condition (red line), and Passive condition (black line) in the overt speech experiment. The motor-corrected waveforms were generated by subtracting the activity generated in the motor-control condition from each participant's Match, Mismatch, and Passive waveforms. Voltage maps are plotted separately for each condition; white dots illustrate the electrodes used in the analysis. (C) Box-and-whiskers plots showing the amplitude of the N1 component elicited by the audible phonemes in the Match, Mismatch, and Passive conditions in the overt speech experiment, using motor-corrected data for the Match and Mismatch conditions. The edges of the boxes represent the top and bottom quartiles, the horizontal stripe represents the median, the cross represents the mean, the whiskers represent the 9th and 91st percentiles, and the colored dots represent the participants whose raw data fell outside the range defined by the whiskers. (D) Scatterplots showing the within-subject difference scores (in terms of N1-amplitude) for the three contrasts-of-interest in the overt speech experiment; namely Match minus Mismatch, Match minus Passive, and Mismatch minus Passive. These difference scores were approximately normally distributed with no clear outliers. Each dot represents a single participant's difference score. The horizontal bars represent the mean, and the error bars represent the 95% confidence interval.

DOI: https://doi.org/10.7554/eLife.28197.010

The following source data is available for figure 5:

**Source data 1.** Overt Speech Experiment - N1 amplitude data.
DOI: https://doi.org/10.7554/eLife.28197.011

---

*Oestreich et al., 2015*; *Houde et al., 2002*). There is strong evidence that the mechanistic basis of N1-suppression to overt speech involves an efference-copy mediated IFM (*Eliades and Wang, 2003*; *Crapse and Sommer, 2008*; *Rauschecker and Scott, 2009*). It thus seems reasonable to assume that a similar mechanism underlies sensory attenuation in the case of inner speech. The idea that inner speech shares mechanistic features with overt speech is consistent with the conceptualization of inner speech '*as a kind of action*' (*Jones and Fernyhough, 2007*, p.396) and, more generally, with Hughlings Jackson's belief that thinking is merely our most complex motor act: '*sensori-motor processes… form the anatomical strata of mental states*' (*Hughlings Jackson, 1958*, p.49). In the words of Oppenheim and Dell (2010, p.1158), '*inner speech cannot be independent of the movements that a person would use to express it*'.

It was notable that inner speech production resulted in N1-suppression if – and only if – the content of the inner phoneme matched the content of the audible phoneme; that is, only in the Match condition. That said, we note that comparisons between the Match/Mismatch conditions on the one hand, and the Passive condition on the other, should be treated with a degree of caution. This is because data for these conditions came from different trial blocks, and (most importantly) differed in terms of the task that participants were required to perform (i.e., produce a covert/overt phoneme at the precise sound-time, versus passively listen to the audible phoneme). Notwithstanding the fact that the N1-suppression effect has previously been found to be robust to variations in attention (*Timm et al., 2013*; *Saupe et al., 2013*) and trial structure (*Baess et al., 2011*), this nevertheless raises the possibility that differences in participants' attention or task-preparation may have contributed to the observed differences between the 'active' conditions and the passive condition. We note that this limitation is not restricted to the current study – it potentially applies to any procedure that attempts to measure sensory suppression by comparing active and passive conditions, which constitutes the vast majority of studies that have examined sensory suppression to overt speech. Critically, however, this limitation does not apply to the key contrast between the Match and Mismatch conditions. This is because data for these conditions came from the same trial blocks, in which participants were required to perform exactly the same task on each trial. For example, in blocks in which participants were required to produce the inner phoneme /ba/, they experienced both Match trials (in which the audible phoneme was /BA/) and Mismatch trials (in which the audible phoneme

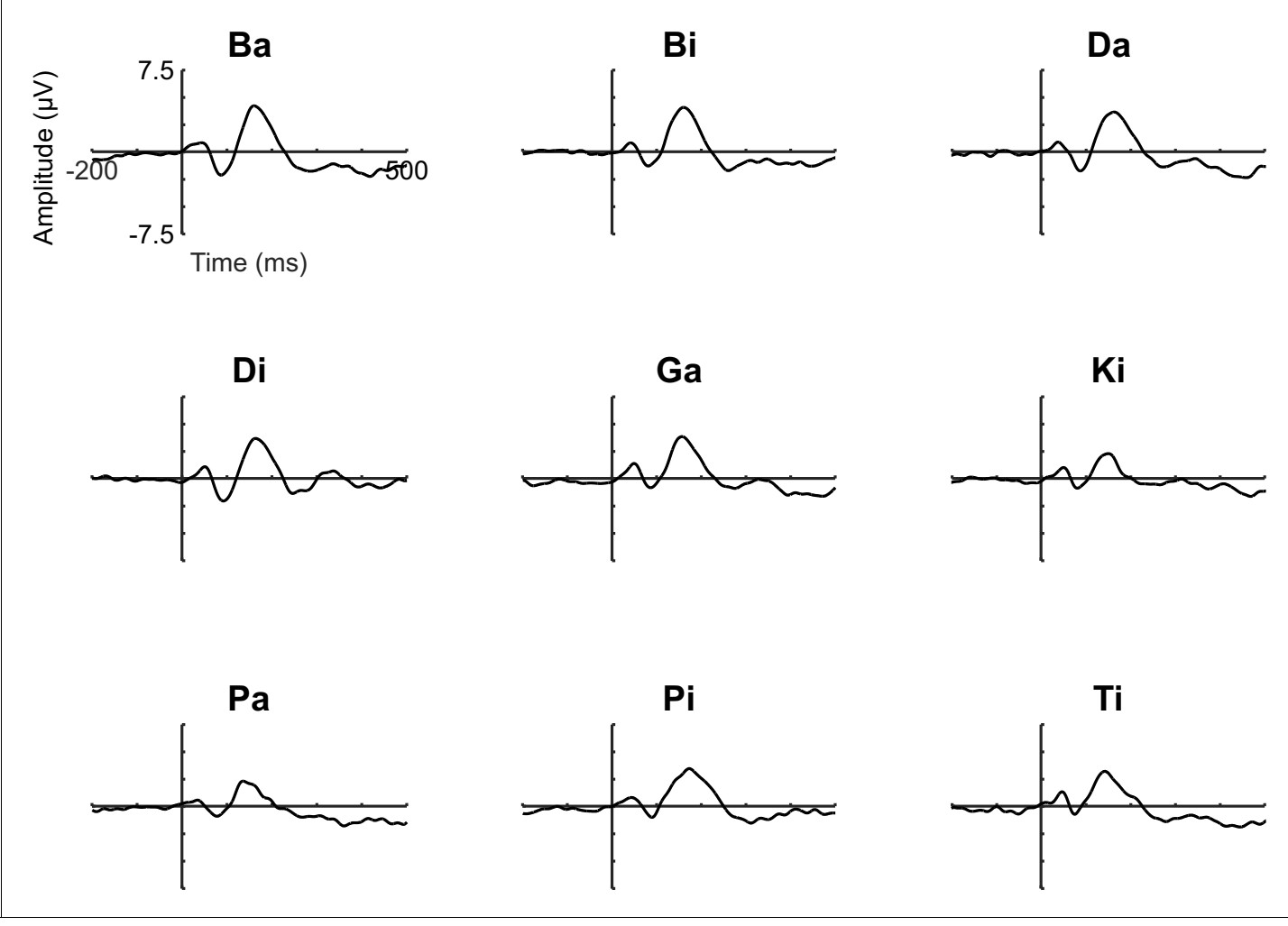

**Figure 6.** Auditory-evoked potentials elicited by nine different phonemes; namely: /BA/, /BI/, /DA/, /DI/, /GA/, /KI/, /PA/, /PI/, and/TI/. Each phoneme was ~200 ms in duration, presented at ~70 dB SPL, and was produced by the same male speaker. Each phoneme was presented 90 times; the presentation order was randomized. Participants were instructed to simply sit quietly and listen to the phonemes. Of the nine different phonemes, /BA/ and /BI/ were judged to be most similar in terms of their amplitude and overall shape, and hence these phonemes were chosen to be used as the audible phonemes in both the inner and overt speech experiments.

DOI: https://doi.org/10.7554/eLife.28197.012

was /BI/), and there was no way for participants to predict whether the current trial would be a Match or Mismatch trial prior to the onset of the audible phoneme. Hence attention, task-preparation, etc., during the pre-stimulus period could not differ systematically between Match and Mismatch trials. Our factorial design also ensured that the differences between the Match and Mismatch conditions were not driven by differences in the inner phoneme (i.e., /ba/ vs /bi/) or differences in the audible phoneme (i.e., /BA/ vs /BI/). Consequently, we can conclude with confidence that the observed difference in N1 amplitude between Match and Mismatch trials reflects the impact of participants' inner speech on their sensory response to the audible phoneme. It is this contrast that demonstrates that inner speech, like overt speech, is associated with a precise, content-specific efference copy.

The results of the inner speech experiment mirror those of previous studies which have found sensory attenuation to *overt* speech to be reduced or eliminated if auditory feedback deviates from what would normally be expected; e.g., by pitch-shifting the feedback or providing foreign-language feedback (*Heinks-Maldonado et al., 2006*; *Behroozmand and Larson, 2011*; *Behroozmand et al., 2011*; *Behroozmand et al., 2016*; *Larson and Robin, 2016*). To confirm this pattern using the

current procedure, we performed a supplementary experiment in which participants were required to overtly vocalize the phoneme at the sound-time. The key result from the overt speech experiment was that N1-amplitude elicited by an audible phoneme was significantly smaller when participants simultaneously produced the same phoneme in their overt speech (Match condition), compared to when they produced a different phoneme in their overt speech (Mismatch condition). This result, which is consistent with numerous previous studies in the sensory attenuation literature (*Heinks-Maldonado et al., 2006*; *Behroozmand and Larson, 2011*; *Behroozmand et al., 2011*; *Behroozmand et al., 2016*; *Larson and Robin, 2016*), was identical to the corresponding contrast in the inner speech experiment, in which participants had to produce phonemes in inner, as opposed to overt, speech.

A second notable finding of the overt speech experiment was that N1-amplitude was significantly smaller in the Passive condition compared to both active conditions (i.e., Match and Mismatch). A potential 'low-level' explanation for this result lies in the fact that participants were required to make an overt motor action in the overt speech experiment but not the inner speech experiment. While the marked difference in pre-stimulus activity between the active and passive conditions (*Figure 5a*) is consistent with this explanation, the fact that between-condition differences remained when correcting for the motor-associated activity (*Figure 5b*) stands against this possibility (though see *Horváth, 2015*; *Sams et al., 2005* for a discussion of the challenges associated with motor correction when comparing active and passive conditions in studies of sensory attenuation). An alternative explanation for why N1 was smallest in the Passive condition is based on the idea that the auditory N1-component is (in addition to sound intensity, discussed previously) also sensitive to stimulus predictability, with predictable sounds evoking a smaller N1 than unpredictable sounds (*Behroozmand and Larson, 2011*; *Bäss et al., 2008*; *Lange, 2011*). Our task differed from other willed vocalization tasks in the literature in that the audible phoneme delivered to the headphones was: (a) of a different person's voice, and (b) much louder than the actual vocalization, as participants were instructed to vocalize quietly in order to minimize bone conduction. In other words, a substantial discrepancy between the predicted and actual sound existed, even in the Match condition. This sensory discrepancy was even larger in the Mismatch condition, as the content of the sound was also different. Consistent with the idea that N1-amplitude is sensitive to stimulus predictability, it is possible that the larger N1-amplitude in the active compared to the passive conditions was due to prediction-errors as to the expected quality of the audible phoneme. It is further possible that such detailed predictions as to phoneme quality do not occur in the context of inner speech (a suggestion for which there is some empirical support – *Oppenheim and Dell, 2008*), which may account for why N1-amplitude was not reduced in the passive condition in the inner speech experiment.

In summary, both the inner speech and overt speech experiments showed the same basic pattern of results with respect to the key contrast: N1-magnitude was smaller if the phoneme generated by the participant (either covertly or overtly) matched the audible phoneme than if it mismatched. These findings suggest that inner speech – like overt speech – is associated with a precise, *content-specific efference copy*, as opposed to a generic and non-specific prediction. Taken together, our results provide support for the contention that inner speech is a special case of overt speech, which does not have an associated motor act. The notion that inner speech generates an IFM in the absence of an overt motor act has been hypothesized previously across several different literatures (*Jones and Fernyhough, 2007*; *Feinberg, 1978*; *Scott, 2013*; *Guenther and Hickok, 2015*; *Ford et al., 2001b*; *Sams et al., 2005*; *Kauramäki et al., 2010*). However, this hypothesis has been notoriously difficult to test empirically, due to the covert nature of inner speech. Ford et al., (*Ford and Mathalon, 2004*) played participants repeated sentences over 30 s and asked them to reproduce the same sentences in inner speech. Ford et al., found that the sentences elicited a smaller N1-component when participants engaged in inner speech compared to when they did not, consistent with the results of the present study. More recently, Tian and Poeppel (*Tian and Poeppel, 2010*) used MEG to show that the auditory cortex was activated immediately following production of an inner phoneme in the absence of auditory feedback, which they took as evidence of an inner-speech-initiated efference copy. In a subsequent study, which was a strong influence on our own, Tian and Poeppel (*Tian and Poeppel, 2013*) asked participants to produce an inner phoneme within a 2.4 s window. This window was followed by an audible phoneme that could either match or mismatch the content of the inner phoneme. The authors found no difference in the amplitude of the M100 between the match and mismatch conditions, inconsistent with the results of the present

study. However, given the width of the temporal window in which participants were asked to produce their inner phoneme (2.4 s), the efference copy and auditory feedback would not necessarily be expected to coincide under these conditions, in which case M100-suppression would not be expected to occur. Tian and Poeppel (*Tian and Poeppel, 2015*) asked participants to signal their production of an inner phoneme via a button-press, and measured the amplitude of the M100 component evoked by a pre-recorded audible phoneme of their own voice which matched the content of the inner phoneme, but which could be pitch-shifted or delayed. They found evidence of M100 suppression to unshifted, undelayed audible phonemes relative to a passive baseline condition, consistent with the results of the present study. Pitch-shifting or delaying the auditory phonemes was found to increase M100 amplitude above baseline levels. While this study's design enabled the timing of the inner phoneme to be precisely specified, the fact that it was specified by means of an overt motor action (i.e., a button-press), which is known to be associated with N1-suppression per se (*Hughes et al., 2013*), raises the possibility of the motor action and inner speech being confounded. Finally, Ylinen et al., (*Ylinen et al., 2015*) asked participants to mentally rehearse tri-syllabic pseudo-words in inner speech. After several mental repetitions, an audible pseudoword was played which had either matching or mismatching beginnings and endings to the rehearsed pseudoword. The results revealed that audible syllables that were concordant with participants' inner speech elicited less MEG activity than discordant syllables, a result which is broadly consistent with the results of the present study.

The current experiment holds some methodological advantages over previous designs. Firstly, the experimental stimuli (animation, audible phonemes, rating-scale, etc.) were physically identical across all conditions, as was the nature of participants' task (i.e., to fixate on the screen and produce an inner phoneme at a designated time). The only thing that differed between the different trial-types was the inner phoneme that participants were asked to produce. This meant that the observed differences in sensory attenuation could not have been due to any physical differences between the conditions (*Luck, 2005*). Secondly, the fact that it was impossible to predict which of the two audible phonemes would be presented on any given trial meant that it was impossible to distinguish Match from Mismatch trials *a priori*. This meant that the observed results could not have been due to between-condition differences in, for example, demand characteristics. Thirdly, the 'ticker tape' feature of the current protocol enabled participants to very accurately time-lock the onset of their inner phoneme to match the onset of the external sound. In the current protocol, the position of the trigger line refreshed every 8.3 ms, which presumably enabled participants to time the onset of their inner phoneme far more accurately than would be possible with a countdown sequence, mental rehearsal or open temporal window, such as have been used in previous studies (*Tian and Poeppel, 2013*; *Ford et al., 2001b*; *Ylinen et al., 2015*). Finally, the current protocol did not require participants to make an active movement to signal the onset of their inner phoneme, such as by pressing a button. This is a significant advantage over previous studies which have employed a motor condition to signal the onset of covert actions, as it avoids the potential confound associated with having temporally-overlapping auditory-evoked and motor-evoked potentials – see Horvath (*Horváth, 2015*) and Neszmélyi and Horvath (*Neszmélyi and Horváth, 2017*) for a discussion of the challenges associated with 'correcting' for motor activity in studies of sensory attenuation. In light of its methodological features, the present study provides arguably the strongest evidence to date that inner speech results in sensory attenuation of the N1-component of the auditory-evoked potential, even in the absence of an overt motor response. Perhaps the most important strength of this paradigm is that all the above issues were controlled within a single task, thereby removing any reliance on cross-experimental inferences.

This study's focus on the N1 component is consistent with the majority of the existing literature on electrophysiological sensory attenuation. The rationale for focusing on N1 lies in the fact that the amplitude of this component is volume dependent; that is, other things being equal, loud sounds evoke N1-components of larger amplitude than do soft sounds (*Näätänen and Picton, 1987*; *Hegerl and Juckel, 1993*). In prior studies of N1-suppression, participants have typically generated sounds through overt actions such as overt speech, button-presses etc. The observation of N1-suppression in such studies thus implies that the brain processes self-generated sounds as though they were physically softer than identical external sounds. The N1-suppression demonstrated in the present study extends this idea by suggesting that the brain also processes *imagined* sounds as though they were physically softer than identical, unimagined sounds. In addition to providing evidence that

inner speech is associated with an IFM of similar nature to overt speech, this finding provides evidence that mental state influences perception at a fundamental level (*Gregory, 1997*).

With regards to the question of what mechanism could underlie sensory attenuation to inner speech: a recent study by Niziolek, Nararajan and Houde (*Niziolek et al., 2013*) on sensory attenuation in the context of *overt* speech production found that the degree of sensory attenuation was stronger when participants produced vowel sounds that were closer (in terms of their acoustic properties) to their median production of these sounds, compared to when they produced vowel sounds that were, for them, less typical. These results suggests that the efference copy associated with overt speech production represents a sensory goal (i.e., *'a prototypical production at the center of a vowel's formant distribution'*, p. 16115), and that the distance (in formant space) of any given utterance from this 'sensory prototype' determines its degree of sensory attenuation. If inner speech is, in fact, a special case of overt speech (as we have suggested above), then this raises the question of the nature of the sensory goal in the context of inner speech production. One possibility, based on the results of Niziolek et al., is that in the present study (in 'imagine /ba/' trials, for example) the sensory goal was of a prototypical /ba/, which was presumably covertly 'spoken' in the participant's own voice (though see below for a discussion of the validity of this assumption). In this case, the participant's prototypical /ba/ would never match perfectly with the audible phoneme, as the audible phoneme would never be the participant's own voice.

The fact that the present study observed N1-suppression in the Match condition but not the Mismatch condition is nevertheless consistent with the *Niziolek et al., 2013* account, in that the distance, in formant space, between an inner /ba/ and an audible /BA/ would be smaller than the distance between an inner /bi/ and an audible /BA/, even though the inner phoneme did not match the audible phoneme perfectly in either case (as the audible phoneme was always produced by the same unknown speaker). The fact that while *Niziolek et al., 2013* observed maximal levels of sensory attenuation to prototypical vowel sounds, they still observed significant (i.e., non-zero) levels of sensory attenuation to atypical vowel sounds is also consistent with this idea. A prediction of this account is that participants should show even greater levels of sensory attenuation in the Match condition if the audible phoneme is presented in their own voice rather than the voice of an unknown stranger; testing this prediction may be a worthwhile endeavor in future studies.

In regards to the assumption that the sensory goals of inner speech are the same as overt speech, and that a person's inner voice is the same as their actual voice, there is some evidence in support of this conjecture: *Filik and Barber (2011)* provided evidence that people produce inner speech in the same regional accent as their overt speech. However, other studies have reported evidence suggesting that inner speech has impoverished acoustic properties relative to overt speech (*Oppenheim and Dell, 2008*). It is also possible that inner speech can consist of several distinct 'voices', with each having specific auditory properties; the fact that people with auditory-verbal hallucinations often report hearing multiple voices with different auditory properties (*McCarthy-Jones et al., 2014*) is consistent with this idea, if – as discussed further below – auditory-verbal hallucinations ultimately reflect inner speech being misperceived as overt speech. Finally, it is also possible that the acoustic properties of inner speech are not fixed. Specifically, in the context of the present study, it is possible that the acoustic properties of the audible phonemes began to influence the inner phonemes, such that after numerous repetitions, participants began to imagine themselves producing an inner phoneme with the acoustic properties of the audible phoneme. Testing these possibilities may also be worthwhile in future studies.

While the primary focus of the paper was on the N1-component of the auditory-evoked potential, between-condition differences were also observed in the amplitude of the P2 and P3 components (see *Figures 3* and *4*). A likely explanation for the observed results in these later components involves another ERP component, the N2, whose spatial and temporal distribution typically overlaps with that of the P2 (*Griffiths et al., 2016*). The N2 and P3 components are among the most heavily investigated components in the ERP literature (*Näätänen and Picton, 1986*; *Polich, 2007*), and are typically elicited by tasks – such as the auditory oddball and Go/NoGo tasks – in which the participant is asked to identify (by means of a button-press, for example) 'target' stimuli which are nested among 'non-target' stimuli (*Smith et al., 2010*; *Spencer et al., 1999*). Critically, the N2 and P3 can also be elicited by tasks in which a mental response is required, such as when target stimuli have to be mentally counted as opposed to signaled with a button-press (*Mertens and Polich, 1997*). We suggest that, in the two inner speech conditions of our study, participants made a mental response –

possibly a 'template-matching response' along the lines of whether the audible phoneme matched their inner phoneme (*Griffiths et al., 2016*) – which they did not make in the Passive condition. In this case, the audible phoneme in the inner speech conditions might be expected to elicit an N2 and a P3 component, which would not be present in the Passive condition. The occurrence of a (negative-going) N2 in the inner speech condition would then interact and compete with the expression of a (positive-going) P2 component elicited by the audible phoneme. The result would be the absence of a distinct P2, but presence of a P3, in the inner speech conditions but not the Passive condition – as observed empirically. It is also worth noting that Tian and Poeppel (*Tian and Poeppel, 2013*) observed a larger M200 component in their match vs. mismatch comparison, consistent with the enhanced P2 observed in the Match vs. Mismatch comparison in the present study. Taken together, these results suggest that the M200/P2 component may index something other than a sensory prediction, possibly involving a cognitive 'template matching' process.

The implied existence of an efference copy to inner speech holds important implications for how to best understand some of the psychotic symptoms associated with schizophrenia. Some of the most characteristic of these symptoms seem to reflect the patient misattributing, to external agents, self-generated motor actions (e.g., delusions of control) and self-generated thoughts (e.g., delusions of thought insertion, auditory-verbal hallucinations – *Feinberg, 1978*; *Frith, 1987*). An influential account of these experiences argues that they arise because of an abnormality in the IFM associated with both *physical* and *mental* actions (*Feinberg, 1978*; *Frith, 1992*). This IFM abnormality leads to an inability to predict and suppress the consequences of self-generated actions, which leads to confusion as to their origins. This hypothesis has a strong theoretical foundation: for example, the distinctive symptom of thought echo, in which the patient hears their own thoughts spoken out loud by an external voice, can be well explained as the patient's own inner speech being misattributed and misperceived as an external voice (*Frith, 1992*). However, while numerous studies have provided empirical evidence showing that schizophrenia patients exhibit subnormal levels of sensory attenuation to their own *physical* actions (*Whitford et al., 2011*; *Blakemore et al., 2000b*; *Shergill et al., 2005*), including subnormal levels of N1-suppression to overt speech (*Ford et al., 2001a*; *Ford et al., 2007b*; *Ford et al., 2001c*), there is little empirical evidence that schizophrenia patients show sensory attenuation deficits to self-generated mental actions such as inner speech. Furthermore, the few studies that did report sensory attenuation deficits to inner speech in patients with schizophrenia did not include a 'mismatch' condition, raising the possibility that these sensory attenuation deficits ultimately reflect attentional deficits in the patient group (*Ford and Mathalon, 2004*; *Ford et al., 2001d*). We suggest that the failure to identify electrophysiological sensory attenuation deficits to inner speech in schizophrenia patients is not because the deficits do not exist, but rather because previous experimental protocols have been insufficiently sensitive to detect them. In order to maximize the chances of detecting sensory attenuation deficits to inner speech in schizophrenia patients, we suggest that future experiments should: (a) ensure that the onset of inner speech is precisely time-locked to the audible sound, without reverting to using a willed action (e.g., a button-press) to signal inner speech-onset, as this could potentially lead to the auditory and motor responses being confounded; (b) limit the content of inner speech to phonemes rather than entire sentences as this enables the onset and content of the inner speech to be more tightly controlled; (c) investigate patients exhibiting those symptoms that seem to most clearly reflect misperceived inner speech (e.g., thought echo, other auditory-verbal hallucinations), rather than grouping patients with different symptom profiles into a clinically-heterogeneous 'schizophrenia' group. By providing an optimized protocol for quantifying N1-suppression to inner speech, our hope is that the present study can provide a methodological framework for identifying and assessing sensory attenuation deficits in inner speech in patients with schizophrenia.

In conclusion, the present study demonstrated that engaging in inner speech resulted in sensory attenuation (specifically, N1-suppression) of the electroencephalographic activity evoked by an audible phoneme, *but only if* the content of inner speech matched the content of the audible phoneme. These results suggest that inner speech evokes an efference-copy-mediated IFM, which is both content-specific and time-locked to the onset of inner speech, which is consistent with the existing literature on sensory attenuation to overt speech. Cumulatively, this implies that inner speech may ultimately be '*a kind of action*', and a special case of overt speech, as long suggested by prominent models of language. Accordingly, these findings not only provide insight into the nature of inner speech, but also provide an experimental framework for investigating sensory attenuation

deficits in inner speech, such as have been proposed to underlie some psychotic symptoms in patients with schizophrenia.

## Materials and methods

### Participants

Forty-two healthy individuals participated in the inner speech experiment. Participants' mean age was 23.4 years (SD = 7.3) and 24 were female. Fifty participants were originally recruited for the study, however eight participants generated ≤ 60 usable epochs in one or more conditions and were excluded from further analysis – see *EEG Processing and Analysis* for further details. The study was conducted at the University of New South Wales (UNSW Sydney; Sydney, Australia), and approved by the UNSW Human Research Ethics Advisory Panel (Psychology).

### Procedure

Participants were seated in a quiet, dimly-lit room, approximately 60 cm from a computer monitor (BenQ XL2420T, 1920 × 1080 pixels, 144 Hz), and were fitted with headphones (AKG K77 Perception) and an EEG recording cap. EEG was recorded with a BioSemi ActiveTwo system from 64 Ag/AgCl active electrodes placed according to the extended 10–20 system. A vertical electro-oculogram was calculated by recording from an electrode placed below the left eye, and subtracting its activity from that of electrode FP1; a horizontal EOG was recorded by placing an electrode on the outer canthus of each eye. We also placed an electrode on the tip of the nose, on the left and right mastoid, and on the masseter muscle to detect jaw movements. During data acquisition, the reference was composed of CMS and DRL sites, and the sampling rate was 2048 Hz.

In regards to the animation that participants viewed on each experimental trial: the ticker tape moved at a constant velocity of 6.5°/s, which meant that it took 3.75 s until the trigger line intersected the fixation line. The ticker tape was marked with labels '3', '2' and '1' that passed the fixation line 3 s, 2 s, and 1 s prior to the trigger line (see *Figure 1b*). The two audible phonemes, /BA/ and /BI/, were selected on the basis of a pilot study which indicated that amongst nine candidate audible phonemes (/BA/, /BI/, /DA/, /DI/, /GA/, /KI/, /PA/, /PI/, /TI/), the two audible phonemes /BA/ and /BI/ produced auditory-evoked potentials that were most similar in terms of their amplitude and overall shape (see *Figure 6*). The two audible phonemes were produced by the same male speaker, and were similar in terms of their loudness (~70 dB SPL) and duration (~200 ms).

There were 60 trials in each trial block. Participants were instructed to fix their gaze on the fixation line on every trial. At the start of each block, participants were told that on every trial of that block they should produce a particular inner phoneme (either /ba/ or /bi/) at the exact moment the trigger line interested the fixation line, or (in Passive blocks) that they should simply listen to the audible sound and not try to imagine anything. Each audible phoneme (/BA/ and /BI/) was presented on 50% of trials within each trial block, and the order was randomized for each participant. This meant that, in those blocks in which participants were instructed to generate a particular inner phoneme (i.e., active blocks), on half of trials their inner phoneme matched the audible phoneme, while on half of trials it mismatched. Following each trial, participants were asked to rate their success in imagining the instructed inner phoneme at the sound-time (or in not imagining anything in the Passive condition). These ratings were made on a scale from 1 ('Not at all successful') to 5 ('Completely successful'), and were reported using the computer keypad. Participants' average ratings were 4.09 out of 5 (SD = 0.65) for Match trials, 4.01 (SD = 0.67) for Mismatch trials, and 4.87 (SD = 0.40) for Passive trials. The order of the trial blocks (imagine /ba/, imagine /bi/, or passive) was randomized for each participant. Each block took approximately 7 min to complete, and was repeated twice over the course of the experiment. Stimulus presentation was controlled by MATLAB (MathWorks, Natick, MA), using Psychophysics Toolbox extensions (*Brainard, 1997*; *Kleiner et al., 2007*).

### EEG processing and analysis

The data pre-processing and analysis was performed in BrainVision Analyzer (Brain Products GmbH, Munich, Germany). The EEG data were re-referenced offline to the nose electrode. Data were first notch filtered (50 Hz) to remove mains artefact, and then band-pass filtered from 0.1 to 30 Hz using a phase-shift free Butterworth filter (48 dB/Oct slope). The filtered data were separated into 800 ms

epochs (200 ms prior to sound onset, 600 ms post-onset), and baseline corrected to the mean voltage from –100 to 0 ms. The epochs were corrected for eye-movement artefacts, using the technique described in (*Gratton et al., 1983*), and any epoch with a signal exceeding a peak-to-peak amplitude of 200 μV for any channel was excluded. To ensure data quality, epochs were classified as unusable and excluded prior to analysis if they failed to meet the above criterion, or if the participant rated their success on the trial as ≤2 out of 5. The remaining usable epochs were included in the analysis and used to make the average waveforms for the three conditions. There were an average of 103.1 (SD = 14.8) usable epochs in the Match condition (of a max. possible 120), 97.0 (SD = 18.3) in the Mismatch condition, and 107.2 (SD = 17.6) in the Passive condition.

The amplitude of the N1-component of the auditory-evoked potential was the primary dependent variable. The N1 peak was identified on each participant's average waveform (*Whitford et al., 2011*; *Ford et al., 2007b*) as the most negative local minimum in the window 25–175 ms post-stimulus onset. The auditory N1-component typically has a fronto-central topography (*Näätänen and Picton, 1987*), which was verified in the current data: N1 was maximal at electrode FCz (see *Figure 2a*). Supplementary analyses were also performed on the P2 and P3 components in the inner speech experiment. As it was not possible to use a peak-detection approach for these components (as not all conditions exhibited a clear P2 and P3 peak), time-windows were identified for the P2 (150–190 ms) and P3 (250–310 ms) components. Average voltage within these time-windows was the dependent variable in these supplementary analyses.

## Statistical analysis

Data were analyzed using repeated-measures ANOVA, with one factor *Condition* (three levels: Match, Mismatch and Passive). In the case of a main effect of *Condition*, contrasts were used to unpack the simple effects. The Greenhouse-Geisser correction was used in the case of a violation in the assumption of sphericity. N1 was maximal at electrode FCz for all three conditions, however in order to improve the reliability of the analysis, the data was averaged across FCz and neighboring electrodes Fz and Cz (*Näätänen and Picton, 1987*; *Woods, 1995*). All of the relevant statistics remained significant when analysis was restricted to electrode FCz.

With regards to the supplementary analyses on the P2 and P3 components: the P2 component (150–190 ms) was maximal at electrode Cz; the data were collapsed across Cz and neighboring electrodes FCz and CPz for the statistical analysis. The P3 component (250–310 ms) was maximal at electrode CPz; the data were collapsed across CPz and neighboring electrodes Cz and Pz for the statistical analysis.

Due to the novelty of the paradigm, it was not possible to obtain precise estimates of the expected effect size. Thus we powered our study to detect a small effect size, based on the heuristic provided by (*Cohen, 1969*); our sample size of 42 provided adequate power (β = 0.8) to detect a small effect size ($\eta_p^2 = 0.04$) at α = 0.05. The power analysis was conducted with G*Power software (version 3.1.9.2; *Faul et al., 2007*). Each experiment was performed only once.

## Additional information

### Competing interests

Anthony WF Harris: Dr Harris has received consultancy fees from Janssen Australia and Lundbeck Australia. He has been on an advisory board for Sumitomo Dainippon Pharma. He has received payments for educational sessions run for Janssen Australia and Lundbeck Australia. He is the chair of One Door Mental Health. The other authors declare that no competing interests exist.

### Funding

| Funder | Grant reference number | Author |
| --- | --- | --- |
| Australian Research Council | DP140104394 | Thomas J Whitford<br>David Luque<br>Mike E Le Pelley |
| National Health and Medical Research Council | APP1069487 | Thomas J Whitford<br>Mike E Le Pelley |

| Australian Research Council | DP170103094 | Thomas J Whitford<br>Anthony W.F. Harris<br>Kevin M Spencer |
| Australian Research Council | DE150100667 | Oren Griffiths |
| National Health and Medical Research Council | APP1090507 | Thomas J Whitford |
| U.S. Department of Veterans Affairs | I01CX001443 | Kevin M Spencer |

The funders had no role in study design, data collection and interpretation, or the decision to submit the work for publication.

## Author contributions

Thomas J Whitford, Conceptualization, Resources, Data curation, Formal analysis, Supervision, Funding acquisition, Investigation, Visualization, Methodology, Writing—original draft, Project administration, Writing—review and editing; Bradley N Jack, Conceptualization, Data curation, Software, Formal analysis, Investigation, Visualization, Methodology, Writing—review and editing; Daniel Pearson, Conceptualization, Data curation, Software, Formal analysis, Investigation, Methodology, Writing—review and editing; Oren Griffiths, David Luque, Conceptualization, Data curation, Formal analysis, Investigation, Methodology, Writing—review and editing; Anthony WF Harris, Conceptualization, Funding acquisition, Investigation, Methodology, Writing—review and editing; Kevin M Spencer, Conceptualization, Supervision, Funding acquisition, Investigation, Methodology, Writing—review and editing; Mike E Le Pelley, Conceptualization, Resources, Data curation, Formal analysis, Supervision, Funding acquisition, Investigation, Methodology, Writing—review and editing

## Author ORCIDs

Thomas J Whitford http://orcid.org/0000-0001-9187-3816
Bradley N Jack http://orcid.org/0000-0003-0523-6656
Daniel Pearson http://orcid.org/0000-0003-1903-4019
Oren Griffiths http://orcid.org/0000-0002-9833-9998
David Luque http://orcid.org/0000-0002-3457-9204
Anthony WF Harris http://orcid.org/0000-0002-8617-4962
Kevin M Spencer http://orcid.org/0000-0002-5500-7627
Mike E Le Pelley http://orcid.org/0000-0002-5145-5502

## Ethics

Human subjects: All participants gave written informed consent to participate and to have their data published in scientific journals. The study was conducted at UNSW Sydney (Sydney, Australia), and approved by the UNSW Human Research Ethics Advisory Panel (Psychology) (File # 2499).

## Decision letter and Author response

Decision letter https://doi.org/10.7554/eLife.28197.021
Author response https://doi.org/10.7554/eLife.28197.022

# Additional files

## Supplementary files

• Supplementary file 1. Syntax used for the analysis of the N1-component in the inner speech experiment (amplitude data). The data were analysed with this syntax using the program IBM SPSS Statistics (v. 23).
DOI: https://doi.org/10.7554/eLife.28197.013

• Transparent reporting form
DOI: https://doi.org/10.7554/eLife.28197.014

## Major datasets

The following dataset was generated:

| Author(s) | Year | Dataset title | Dataset URL | Database, license, and accessibility information |
|---|---|---|---|---|
| Whitford T, Jack B, Pearson D, Griffiths O, Luque D, Harris A, Spencer K, Le Pelley M | 2017 | Neurophysiological evidence of efference copies to inner speech | https://osf.io/b58qa/ | Available at OSF Home (htts://osf.io/) |

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
