## [Decision Letter]

Thank you for submitting your article "Neurophysiological evidence of efference copies to inner speech" for consideration by *eLife*. Your article has been reviewed by two peer reviewers, and the evaluation has been overseen by Richard Ivry, serving here as both Senior Editor and Reviewing Editor. The following individual involved in review of your submission has agreed to reveal his identity: Chi-Ming Chen (Reviewer #1).

We have discussed the reviews with one another and I have drafted this decision to help you prepare a revised submission.

Summary:

The study employed electrophysiological methods to explore how the brain responds to covert actions, drawing on a substantial literature showing that the sensory response to self-produced movements is attenuated, presumably due to efference copy signals. The focus here is on an early ERP response, the N100. In a clever design, participants are presented with a visual cue that alerts them to an auditory presentation of a CV sound (/bi/ or /ba/). They are instructed to covertly produce either a /bi/ or a /ba/ (depending on the block) via a moving visual cue that enabled the temporal alignment of the inner speech with the external stimulus. As such, they create conditions in which the covert speech either matches or does not match the stimulus. The results show a reduced N1 amplitude for the match conditions compared to both the mismatch and a control, passive condition. A supplemental experiment using overt speech shows a reduced N1 amplitude for the match condition compared to the mismatch condition. These results make a compelling case for the operation of an efference copy process during inner speech.

Major comments for revisions:

1) There was concern about your use of a blocked design, with all of the reviewers noting that there are multiple advantages to a mixed design. In the end, we decided that this was not of critical concern, especially since the focus here is on the N1; factors such as a shift in attentional set might be more likely to impact the latter components. This could, in part explain the >200 ms results, but could also affect global amplitude/power differences, and these could potentially lead to spurious results in the N1 analysis. One way of perhaps mitigating these concerns would be to assess amplitude/power differences outside of the epochs of interest (e.g. during the baseline) and see if there are significant differences between conditions (i.e. the passive and the match/mismatch). This also provides a check on whether there may be any issues with the block design contributing to the N1 results, something we don't anticipate to be a problem, but would like to have assessed.

2) "Figure 2 shows the auditory-evoked potentials averaged across electrodes FCz, Fz and Cz, as these were the electrodes at which N1 was maximal (see Figure 2, voltage maps)." The voltage montage maps clear show the negative/blue/N1 maximal at the left hemisphere not at the averaged midline electrodes, Fz, FCz, and Cz. This is especially evident in the Match condition, the negative/blue/N1 maximal is clearly at C3 location. Please clarify on this issue.

3) Related to this, you have written, "The auditory N1-component typically has a fronto-central topography [Ford et al., 2014], which was verified in the current data: N1 was maximal at electrode FCz (see Figure 2)." However, there is no Figure 2 in this version of the manuscript. It may be that the current Figure 2's voltage maps were not updated from a previous version (e.g., these maps were from data that were not re-referenced by nose electrode)? However, if the maps are correct, the statements in the manuscript must be wrong and it would not justify using the averaged ERP from Fz, FCz, and Cz.

4) It's hard to tell to what extent this is testing the production of an efference copy since the presented stimuli are not the subjects' own voice. While you refer to "content of the audible phoneme" (Discussion, third paragraph), little effort is made to unpack this statement. We imagine, you are arguing that the matching component of the efference copy works in some sort of abstract space (i.e. comparing the phonemic representation rather than say, the motor or sensory goal), but this seems not to be the case in the literature (see Niziolek et al. 2013). This point requires further commentary. What is the actual mechanism at work here? What is the mapping between an external stimulus and an efference copy when you attempt to compare someone else's speech with your own self-produced speech?

5) While we appreciate the focus on the internal speech experiment, we felt that the control experiment with overt speech provided a nice point of contrast. As things now stand, this experiment is buried in the Discussion. Knowing that some readers never make it that far, we would like the overt speech experiment presented in the Results section.

---

## [Author Response]

Major comments for revisions:1) There was concern about your use of a blocked design, with all of the reviewers noting that there are multiple advantages to a mixed design. In the end, we decided that this was not of critical concern, especially since the focus here is on the N1; factors such as a shift in attentional set might be more likely to impact the latter components. This could, in part explain the >200 ms results, but could also affect global amplitude/power differences, and these could potentially lead to spurious results in the N1 analysis. One way of perhaps mitigating these concerns would be to assess amplitude/power differences outside of the epochs of interest (e.g. during the baseline) and see if there are significant differences between conditions (i.e. the passive and the match/mismatch). This also provides a check on whether there may be any issues with the block design contributing to the N1 results, something we don't anticipate to be a problem, but would like to have assessed.

Thank you for raising this point. We designed the experiment such that while participants were required to perform the same mental action on every trial in each (60-trial) block, the audible phoneme presented to their headphones (i.e., either /BA/ or /BI/) was unpredictable for any given trial. The upshot of this design is that it meant that while the Passive trials were effectively ‘blocked’, the Match and Mismatch trials were ‘mixed’, and thus any differences between the Match and Mismatch conditions could not have been due to extraneous factors such as between-condition differences in attention (as highlighted in of the previous version of the manuscript). This is critical, because the Match versus Mismatch comparison constituted the key contrast around which the vast majority of the Discussion was framed: it is this contrast which demonstrates most clearly that inner speech is associated with a precise, content-specific efference copy.

Our rationale for designing the experiment in this way was that we were concerned a fully-mixed design would be confusing to participants, as it would require them receiving different instructions as to what mental action they should perform on each individual trial. We were concerned that having instructions that changed every trial could lead to participants forgetting or ignoring the instructions, and/or limit participants’ ability to produce the inner phonemes clearly. However, we agree that our design could potentially have led to unanticipated differences between the Passive condition and the two inner speech conditions (i.e., Match and Mismatch), possibly due to factors such as attention.

As per your suggestion, one way of mitigating this concern would be to compare the Passive condition with the inner speech conditions across the pre-stimulus period. As illustrated by the overlapping waveforms in the baseline period in Figure 2, there were no significant differences between the three conditions at any point in the original -200 to 0 ms baseline (which was baseline corrected to the average voltage in the period -100 to 0 ms). However, in order to gain a more global perspective of pre-stimulus activity, we extended the pre-stimulus period all the way out to -3000 ms (i.e., 3000 ms pre-stimulus), and baseline corrected to the average voltage in the period -3000 to -2500 ms. While the waveforms for the three conditions largely overlapped for the first ~1900 ms of the extended epoch, at around 1100 ms pre-stimulus, the waveforms for the two inner-speech conditions (i.e., Match and Mismatch) began to diverge from the Passive waveform, and began to exhibit a slow, negative-going deflection – see Author response image 1, left hand panel.

We believe that this deflection may reflect a readiness-potential (RP) that is associated with the participant preparing to generate the inner phoneme. The basis for this assertion is that the negative deflection we observed is similar – in terms of its global form, onset, and scalp distribution – to the negative deflection which has previously been reported in studies of *overt* speech production, and which has been previously argued to reflect an RP (e.g., see McArdle et al., 2014, Clinical Neurophysiology, 120, 275-284; Jansen et al., 2014, Journal of Neuroscience Methods, 232, 24-29; Wolhert, 1993, Journal of Speech and Hearing Research, 36, 897-905; Deecke et al., 1986, Experimental Brain Research, 65, 219-223). Specifically, the negative deflection we observed had the same basic form (i.e., a slow, monotonic, negatively-going waveform), a similar onset (1000+ ms pre-stimulus), and a similar topography (a prominent central negativity) as the negative deflection reported in these previous studies of overt speech production. Furthermore, we also observed a similar negative deflection in the extended pre-stimulus period in our overt speech experiment, which directly replicated the results of the aforementioned studies. The negative deflection we observed in the overt speech experiment had a similar form and topography to the deflection observed in the inner speech experiment (compare the left and right hand panels in Author response image 1) though it had a slightly delayed onset relative to the inner speech experiment, and was of larger magnitude (note the different scales between the panels).

The identification of an RP to inner speech – a purely mental action – would be noteworthy, given that this component has traditionally been assumed to require an *overt* motor action. Identifying an RP to inner speech would also provide further evidence for our paper’s central tenet, which is that inner speech is ultimately processed by the brain *“as a kind of action”*, and may in fact reflect a special case of overt speech.

However, we urge caution in making this interpretation, as it is not possible to definitely determine whether the negative deflection observed in the current data is, in fact, an RP. Specifically, while the observed negative deflection could well reflect an RP, it could also potentially reflect a Contingent Negative Variation (CNV), or perhaps even a combination of these two components. The CNV is a slow, negative-going waveform that is elicited when “a warning stimulus (S1) announces that, within a few seconds, an imperative stimulus (S2) will arrive, asking for a quick response” (Brunia et al., 2012, p. 189). On the face of it, the ticker tape feature of the present design, which announced the impending arrival of the imperative trigger line, would seem to provide fertile ground for the generation of a CNV. Furthermore, the RP to overt speech and the CNV are known to have a similar global form (i.e., a slow, negative-going waveform in the pre-stimulus period), a similar onset (i.e., 1000+ ms, pre-stimulus), and a similar scalp distribution (i.e., a bilateral central topography). The upshot of this is that determining the relative contributions of the RP and CNV to the negative deflection observed in the present study is not feasible, post hoc. For this reason – and also so as not to distract from the paper’s central finding, namely N1-suppression to inner speech – we have decided not to include a description of the pre-stimulus activity in the extended baseline period in the revised manuscript. However, we would, of course, be happy to add these data to the revised manuscript, if you think they would be of interest to readers.

A crucial point to note is that regardless of whether the negative deflection reflected an RP, a CNV, or a combination of both components, it is almost certainly *not* responsible for the between-condition differences in N1-amplitude we observed. The reason we say this is that we were able to remove the negative deflection in the Match and Mismatch conditions by applying a 1 Hz high-pass filter (phase-shift-free Butterworth filter, 12 dB/octave slope), as can been seen below in Author response image 2, by comparing the left hand (unfiltered) and right hand (filtered) panels.

**Author response image 2. respfig2:** 

Critically, the high-pass filter did not alter N1 amplitude for any condition in any appreciable way, and nor did it alter the pattern of results, as can be seen in Author response image 3 by comparing the left and right hand panels. Thus, we are confident that the observed between-condition differences in N1 amplitude were not driven by between-condition differences in the pre-stimulus period.

**Author response image 3. respfig3:** 

An important point to note here, which is particularly relevant to the issues discussed in the opening paragraph, is that the negative deflection in the pre-stimulus period is far more likely to be due to an RP/CNV in the ‘active’ conditions, as opposed to between-condition differences in block-type. This is because the negative deflection in the pre-stimulus period was also observed in the Motor-Control condition in the overt speech experiment; that is, a condition in which participants vocalized a phoneme at the sound-time, but no audible phoneme was delivered. This was in contrast to the Passive condition in which, to reiterate, the negative deflection did not occur. The significance of this point is that both the Motor-Control and Passive conditions were fully blocked. The fact that an RP was present in the Motor-Control condition (which was blocked), but not present in the Passive condition (which was also blocked), suggests that the observed between-condition differences in pre-stimulus activity were not due to the differences in block-type, but were rather due to the electrophysiological consequences of action preparation (both mental [inner speech] and physical [overt speech]), and/or stimulus anticipation. Finally, we would again emphasize that the Passive condition is in some ways fundamentally different from both the Match and Mismatch conditions (for both the inner and overt speech experiments), in that the latter conditions involved the performance of a mental or physical ‘action’ (i.e., the production of an inner or overt phoneme), while the Passive condition did not. The Passive condition can thus only ever provide a rough ‘point-of-reference’ for the active conditions. The key comparison, we emphasize, was between the Match and Mismatch conditions, and these conditions were matched in terms of both the active processes involved (i.e., both involved the same mental or physical action), and how the trials were presented (i.e., a mixed design).

We now discuss the potential issues associated with between-condition differences in block-type explicitly, in the following additions to the manuscript text:

[Results, footnote]: The decision was taken to block trials in this way in order to make the task more ergonomic for participants, by having them perform the same task (imagine /ba/, imagine /bi/, or passively listen) on every trial of a block. […] Importantly, however, our procedure meant that the critical Match and Mismatch trials were fully and unpredictably intermixed in those blocks in which participants were required to produce an inner phoneme. We return to this issue in the Discussion.

[Discussion]: That said, we note that comparisons between the Match/Mismatch conditions on the one hand, and the Passive condition on the other, should be treated with a degree of caution. […] It is this contrast that demonstrates that inner speech, like overt speech, is associated with a precise, content-specific efference copy.

[Discussion footnote]: We note that this limitation is not restricted to the current study – it potentially applies to any procedure that attempts to measure sensory suppression by comparing active and passive conditions, which constitutes the vast majority of studies that have previously examined sensory suppression to overt speech.

2) "Figure 2 shows the auditory-evoked potentials averaged across electrodes FCz, Fz and Cz, as these were the electrodes at which N1 was maximal (see Figure 2, voltage maps)." The voltage montage maps clear show the negative/blue/N1 maximal at the left hemisphere not at the averaged midline electrodes, Fz, FCz, and Cz. This is especially evident in the Match condition, the negative/blue/N1 maximal is clearly at C3 location. Please clarify on this issue.

Thank you for identifying this apparent contradiction. When preparing our response to this point, we began by checking the N1-amplitudes at all of the electrodes, and found that the N1-amplitude was indeed maximal at electrode FCz for all three conditions; i.e., consistent with what we claimed in the text, but inconsistent with the voltage maps. However, we then realized the source of our error: while the N1 amplitude and latency data described in the manuscript (on which the statistics were based) were calculated from participants’ peak-picked data, the voltage maps were calculated based on a time-window around the N1-peak on the grand-average waveform. In other words, the data on which the statistics were calculated were not the same data from which the voltage maps were generated. We thus modified the N1 voltage maps so that they were calculated based on the same data that were analyzed statistically (i.e., participants’ peak-picked data). The revised voltage maps showed the expected pattern: that is, N1 was maximal at FCz in all three conditions – see the revised voltage maps in Figure 2.

We repeated this approach for the N1-analysis in the overt speech experiment and again, the results revealed a maximum at in all three conditions – see the revised voltage maps in Figure 3.

As can be seen in the revised Figure 2 (top), the revised topographies retained a slight leftwards-shift in the inner speech experiment, at least for the Match condition. Thus in order to ensure the stability of the results, we re-ran the analyses with an expanded set of electrodes to ensure that we captured the region in which N1 was maximal: specifically, in addition to the three midline electrodes (Fz, FCz, Cz), we entered the three electrodes that were immediately to the left (F1, FC1, C1) and right (F2, FC2, C2) of these midline electrodes. The observed pattern of results remained identical when using this expanded set of electrodes. This supplementary analysis is now included in the Results section of the revised manuscript:

[Results]: As can be seen in Figure 2, while the topographies exhibited a fronto-central negativity in all three conditions, centered on electrode FCz, there was a hint of a leftward shift in the scalp distribution in the Match condition. […] The difference between the Mismatch and Passive conditions remained non-significant (t(41) = 0.11, p =.916, dz = 0.02, CI(95%) = [-0.889, 0.800]. The Condition × Electrode interaction was also not significant (F(16,656) = 1.18, p =.323, η_p_^2^ = 0.028).

3) Related to this, you have written, "The auditory N1-component typically has a fronto-central topography [Ford et al., 2014], which was verified in the current data: N1 was maximal at electrode FCz (see Figure 2)." However, there is no Figure 2 in this version of the manuscript. It may be that the current Figure 2's voltage maps were not updated from a previous version (e.g., these maps were from data that were not re-referenced by nose electrode)? However, if the maps are correct, the statements in the manuscript must be wrong and it would not justify using the averaged ERP from Fz, FCz, and Cz.

As discussed in our response to the previous point, we can confirm that the N1 component was indeed maximal at FCz for all three conditions. However, as discussed above, the voltage maps in Figure 2 were incorrect in the original manuscript, as they were based on a time-window taken from the grand-average waveform, as opposed the peak-picked data on which the analyses were based. We have rectified this issue in the revised manuscript and now present the correct topographies.

4) It's hard to tell to what extent this is testing the production of an efference copy since the presented stimuli are not the subjects' own voice. While you refer to "content of the audible phoneme" (Discussion, third paragraph), little effort is made to unpack this statement. We imagine, you are arguing that the matching component of the efference copy works in some sort of abstract space (i.e. comparing the phonemic representation rather than say, the motor or sensory goal), but this seems not to be the case in the literature (see Niziolek et al. 2013). This point requires further commentary. What is the actual mechanism at work here? What is the mapping between an external stimulus and an efference copy when you attempt to compare someone else's speech with your own self-produced speech?

Thank you for raising this interesting and important issue. The study of Niziolek et al. (2013) provided evidence that the efference copy associated with the overt production of speech (in this case, a vowel sound) predicted *“*a prototypical production at the center of a vowel’s formant distribution”, p. 16115). Furthermore, they found that the degree of sensory attenuation was related to a spoken sound’s distance (i.e., in formant space) from this prototype; i.e., sounds closer to an individual’s median production were attenuated more than sounds further away.

If inner speech is, in fact, a special case of overt speech – as we have suggested in the manuscript – then this raises a question as to the nature of the sensory prototype to inner speech. Specifically, with regards to the present study, it raises the question as to why the Match condition showed sensory attenuation, given that the audible phoneme was not presented in the participant’s own voice, and thus would could never perfectly match the sensory prototype. (This argument assumes that the sensory prototype of inner speech has the auditory properties of the participant’s own voice; we discuss this issue further below).

We have attempted to address these important theoretical issues in detail in the following addition to the Discussion section. In a nutshell, we suggest that if the sensory prototype is the participant’s own voice, then while the inner phoneme in the Match condition would not match this sensory prototype perfectly, it would be a *better* match than the inner phoneme in the Mismatch condition, and would thus be expected to be associated greater – though not maximal – levels of sensory attenuation. We then go on to discuss the possibility that the assumption that the sensory prototype is of the participant’s own spoken voice may not, in fact, be correct (or at least may not be correct all of the time).

The full addition to the Discussion is as follows:

[Discussion]: With regards to the question of what mechanism could underlie sensory attenuation to inner speech: a recent study by Niziolek, Nararajan and Houde [Niziolek, Nararajan and Houde, 2013] on sensory attenuation in the context of overt speech production found that the degree of sensory attenuation was stronger when participants produced vowel sounds that were closer (in terms of their acoustic properties) to their median production of these sounds, compared to when they produced vowel sounds that were, for them, less typical. […] Testing these possibilities may be also be worthwhile in future studies.

5) While we appreciate the focus on the internal speech experiment, we felt that the control experiment with overt speech provided a nice point of contrast. As things now stand, this experiment is buried in the Discussion. Knowing that some readers never make it that far, we would like the overt speech experiment presented in the Results section.

We agree, and as per your instructions have moved the results of the overt speech experiment from the Discussion to the Results section, and have modified the relevant section in the Discussion where we discuss the implications of the overt speech experiment.